# STYLEADAPTER: A UNIFIED STYLIZED IMAGE GENERATION MODEL WITHOUT TEST-TIME FINE-TUNING

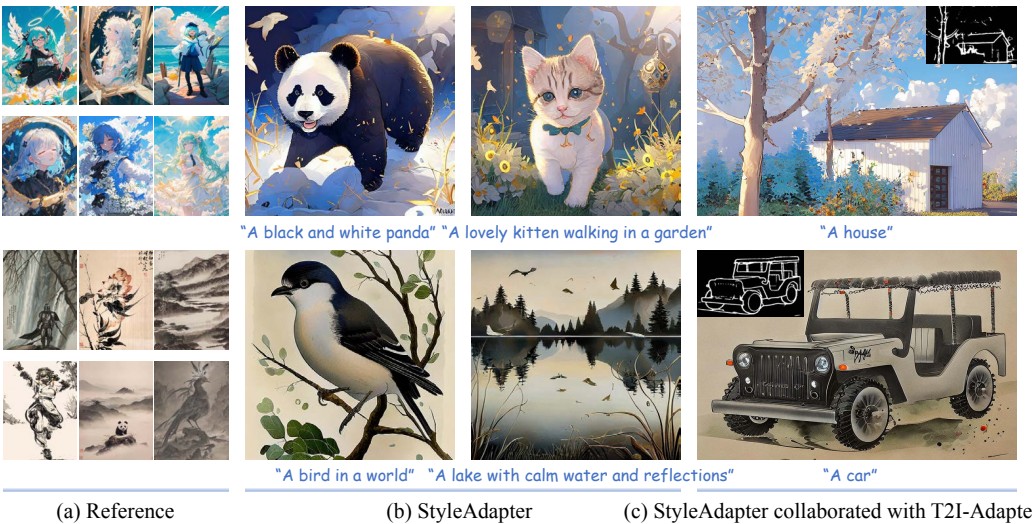

"A black and white panda"  "A lovely kitten walking in a garden"    "A house"

"A bird in a world"  "A lake with calm water and reflections"    "A car"

(a) Reference          (b) StyleAdapter          (c) StyleAdapter collaborated with T2I-Adapter

Figure 1: Given multiple style reference images, our **StyleAdapter** is capable of generating images that adhere to both style and prompts without test-time fine-tuning. Moreover, our method shows compatibility with additional controllable conditions, such as sketches.

## ABSTRACT

This work focuses on generating high-quality images with specific style of reference images and content of provided textual descriptions. Current leading algorithms, i.e., DreamBooth and LoRA, require fine-tuning for each style, leading to time-consuming and computationally expensive processes. In this work, we propose StyleAdapter, a unified stylized image generation model capable of producing a variety of stylized images that match both the content of a given prompt and the style of reference images, without the need for test-time fine-tuning. It introduces a two-path cross-attention (TPCA) module to separately process style information and textual prompt, which cooperate with a semantic suppressing vision model (SSVM) to suppress the semantic content of style images. In this way, it can ensure the controllability of the prompt over the content of the generated images while mitigating the negative impact of semantic information in style references. Besides, our StyleAdapter can be integrated with existing controllable synthesis methods, such as T2I-adapter and ControlNet, to attain a more controllable and stable generation process. Extensive experiments demonstrate the superiority of our method over previous works.

## 1 INTRODUCTION

Recent advancements in data and large-scale models have significantly contributed to the progress made in text-to-image (T2I) generation (Ding et al., 2021; Nichol et al., 2022; Ramesh et al., 2022; 2021; Rombach et al., 2022; Saharia et al., 2022; Zhou et al., 2021). These models are capable of generating high-quality images based on provided prompts. Furthermore, T2I methods can incorporate specific styles into the generated images by using textual descriptions of the style as prompts. However, textual descriptions often lack expressiveness and informativeness compared to

visual representations of styles, resulting in T2I outputs with coarse and less detailed style features. To leverage the rich information present in visual data of styles, previous works (Gal et al., 2022; Zhang et al., 2022b) have proposed textual inversion methods that map visual representations of styles to textual space. This approach enables the style information extracted from visual images to guide T2I models. Nevertheless, these methods still face limitations, as the visual-to-textual projection fails to preserve the rich details inherent in visual images, leading to suboptimal styles in the generated images. Currently, DreamBooth (Ruiz et al., 2022) and LoRA (Hu et al., 2021) offer more effective solutions by employing fine-tuning to the original diffusion model or utilizing extra small networks to adapt to specific styles. These approaches enable the generation of images with relatively precise styles, capturing details such as brushstrokes and textures. However, the need to fine-tune or re-train the model for each new style makes these methods computationally demanding and time-consuming, rendering them impractical for many applications.

Developing a unified model capable of generating various stylized images without test-time fine-tuning is highly desirable for increased efficiency and flexibility. This work aims to propose such a unified model to generate high-quality stylized images that match the content of a given prompt and the style of the style references. However, accurately extracting style information from style images and ensuring that the style information and textual prompts precisely focus on stylization and content generation, respectively, remains a significant challenge. Our vanilla approach reveals that simply extracting style reference features with CLIP's (Radford et al., 2021) vision model and combining them with prompt features as the condition for Stable Diffusion (SD) (Rombach et al., 2022) leads to two main issues: *1) loss of prompt controllability over generated content, and 2) inheritance of both semantic and style features from style references, compromising content fidelity.*

Our in-depth observations and analyses demonstrate that separately injecting contextual prompt and semantic-suppressed style reference information into generated images can effectively ensure prompt controllability and mitigate the negative impact of semantic information in style references. Based on these analyses, we propose StyleAdapter, a unified stylized image generation model that produces a variety of stylized images matching both the content of a given prompt and the style of reference images without test-time fine-tuning. It introduces a two-path cross-attention (TPCA) module to separately process style information and textual prompt, cooperating with a semantic suppressing vision model (SSVM) to suppress style image semantics. This ensures prompt controllability over generated content while mitigating the negative impact of semantic information in style references. Furthermore, StyleAdapter can be integrated with existing controllable synthesis methods, such as T2I-adapter (Mou et al., 2023) and ControlNet (Zhang & Agrawala, 2023), for a more controllable and stable generation process.

Our contributions can be summarized as follows: **(1).** We propose StyleAdapter, a unified stylized image generation model capable of producing a variety of stylized images that match both the content of a given prompt and the style of reference images, without requiring test-time fine-tuning. **(2).** Based on in-depth observations and analyses, we introduce a two-path cross-attention (TPCA) module to separately process style information and textual prompts, which cooperates with a semantic suppressing vision model (SSVM) to suppress the semantic content of style images. It ensures the controllability of the prompt over the generated content while mitigating the negative impact of semantic information in style references. **(3).** Our StyleAdapter can be integrated with existing controllable synthesis methods to generate high-quality images in a more controllable and stable manner.

## 2 RELATED WORKS

### 2.1 TEXT-TO-IMAGE SYNTHESIS

Text-to-image synthesis (T2I) is a challenging and active research area that aims to generate realistic images from natural language text descriptions. Generative adversarial networks (GANs) are one of the most popular approaches for T2I synthesis, as they can produce high-fidelity images that match the text descriptions (Reed et al., 2016; Zhang et al., 2017; 2018; Xu et al., 2018; Li et al., 2019). However, GANs suffer from training instability and mode collapse issues (Brock et al., 2018; Dhariwal & Nichol, 2021; Ho et al., 2022). Recently, diffusion models have shown great success in image generation (Song et al., 2020; Ho et al., 2020; Nichol & Dhariwal, 2021; Dhariwal & Nichol, 2021), surpassing GANs in fidelity and diversity. Many recent diffusion methods have also focused

on the task of T2I generation. For example, Glide (Nichol et al., 2022) proposed to incorporate the text feature into transformer blocks in the denoising process. Subsequently, DALL-E (Ramesh et al., 2021), Cogview (Ding et al., 2021), Make-a-scene (Gafni et al., 2022), Stable Diffusion (Rombach et al., 2022), and Imagen (Saharia et al., 2021) significantly improved the performance in T2I generation. To enhance the controllability of the generation results, ControlNet (Zhang & Agrawala, 2023) and T2I-Adapter (Mou et al., 2023) have both implemented an additional condition network in conjunction with stable diffusion. This allows for the synthesis of images that adhere to both the text and condition.

## 2.2 STYLIZED IMAGE GENERATION

Image style transfer is a task that involves generating artistic images guided by an input image. Traditional style transfer methods match patches between content and style images using low-level hand-crafted features (Wang et al., 2004; Zhang et al., 2013). With the rapid development of deep learning, deep convolutional neural networks have been employed to extract the statistical distribution of features that effectively capture style patterns (Gatys et al., 2016; 2017; Kolkin et al., 2019). In addition to CNNs, visual transformers have also been utilized for style transfer tasks (Wu et al., 2021; Deng et al., 2022). Recently, benefiting from the success of diffusion models (Rombach et al., 2022; Saharia et al., 2021; Ramesh et al., 2021), InST (Zhang et al., 2022b) adapted diffusion models as a backbone to be inverted and as a generator for stylized image generation. Textual inversion (Gal et al., 2022), DreamBooth (Ruiz et al., 2022), LoRA (Hu et al., 2021), and StyleDrop (Sohn et al., 2023) propose fine-tuning the SD for specific concepts or styles. Although these methods are effective, they require fine-tuning the SD model for each concept or style. In contrast, our StyleAdapter aims to generate various stylized images with a unified model without the need for test-time fine-tuning.

## 3 METHODOLOGY

This work aims to propose a unified stylized image generation model capable of producing a variety of stylized images that match both the content of a given prompt and the style of reference images, without the need for test-time fine-tuning. This work builds upon SD. In this section, we first briefly recap SD and a vision model in CLIP (Radford et al., 2021) commonly used to extract vision features from vision data. Then, we introduce a vanilla StyleAdapter, which highlights the challenges in constructing a unified stylized image generation model. Based on in-depth observations and analyses, we propose our delicate StyleAdapter, with a two-path cross-attention (TPCA) module used to separately process style information and textual prompts, and a semantic suppressing vision model (SSVM) used to suppress style image semantics. This approach ensures prompt controllability over generated content while mitigating the negative impact of semantic information in style references.

## 3.1 PRELIMINARY

**Stable Diffusion.** SD is a latent diffusion model (LDM) (Rombach et al., 2022) trained on large-scale data. LDM is a generative model that can synthesize high-quality images from Gaussian noise by iterative sampling. Compared to the traditional diffusion model, its diffusion process happens in the latent space. Therefore, except for a diffusion model, an autoencoder consisting of an encoder $\mathcal{E}(\cdot)$ and a decoder $\mathcal{D}(\cdot)$ is needed. $\mathcal{E}(\cdot)$ is used to encode an image $I$ into the latent space $z$ ($z = \mathcal{E}(I)$) while $\mathcal{D}(\cdot)$ is used to decode the feature in the latent space back to an image. The diffusion model contains a forward process and a reverse process. Its denoising model $\epsilon_\theta(\cdot)$ is implemented with UNet (Ronneberger et al., 2015) and trained with a simple mean-squared loss:

$$L_{LDM} := \mathbb{E}_{z \sim \mathcal{E}(I), c, \epsilon \sim \mathcal{N}(0,1), t} \left[ \| \epsilon - \epsilon_\theta \left( z_t, t, c \right) \|_2^2 \right], \tag{1}$$

where $\epsilon$ is the unscaled noise, $t$ is the sampling step, $z_t$ is latent noise at step $t$, and $c$ is the condition. While SD acts as a T2I model, $c$ is the text feature $f_t$ of a natural language prompt encoded with the text model of CLIP (Radford et al., 2021). $f_t$ is then integrated into SD with a cross-attention model, whose query $\mathbf{Q}_t$ is from the spatial feature $y$ which is extracted from $Z_t$, and key $\mathbf{K}_t$ and value $\mathbf{V}_t$ are from $f_t$. The process can be expressed as:

$$\begin{cases} \mathbf{Q}_t = \mathbf{W}_{Qt} \cdot y; \ \mathbf{K}_t = \mathbf{W}_{Kt} \cdot f_t; \ \mathbf{V}_t = \mathbf{W}_{Vt} \cdot f_t; \\ Attention(\mathbf{Q}_t, \mathbf{K}_t, \mathbf{V}_t) = softmax(\frac{\mathbf{Q}_t \mathbf{K}_t^T}{\sqrt{d}}) \cdot \mathbf{V}_t, \end{cases} \tag{2}$$

where $\mathbf{W}_{Q_t/K_t/V_t}$ are learnable weights, and $d$ is dependent on the number of channels of $y$.

**Vision Model.** The vision model (VM) in CLIP (Radford et al., 2021) are commonly used for extracting feature from vision data in T2I models. To process a vision image, such as our style reference $I_r \in \mathbb{R}^{(H \times W \times C)}$ ($H, W, C$ are the hight, width, and channels, respectively), VM takes a sequence of its flattened patches $I_r^p \in \mathbb{R}^{N \times (P^2 \cdot C)}$ ($P$ is the patch size and $N = HW/P^2$ is the sequence length) as input, and deploys a vision embedding module to attain their embeddings with a linear projection $\mathbf{E} \in \mathbb{R}^{P^2 \cdot C \times D}$. An additive class embedding $E_{cls} \in \mathbb{R}^{1 \times D}$ is attached to the vision embeddings before adding with position embedding $E_{pos} \in \mathbb{R}^{(N+1) \times D}$. The embedding process can be formulated as:

$$E_{Ir} = [E_{cls}, I_r^0 \mathbf{E}, I_r^1 \mathbf{E}, I_r^{N-1} \mathbf{E}] + E_{pos}. \tag{3}$$

Then the $E_{Ir}$ is encoded into vision features $f_r$ with a vision encoder.

## 3.2 Vanilla StyleAdapter with In-Depth Analyses

A straightforward approach to adapt SD for stylized image generation involves extracting the style feature $f_r$ from a style reference $I_r$ using VM and concatenating it with the prompt feature ($f_t$). This concatenated result serves as the condition guiding SD's generation. To enhance the expressiveness of the style feature, we employ an additive style embedding module (**StyEmb**) with a transformer block to embed $f_r$ into $f_s$. As illustrated in Figure 2, StyEmb predefines a learnable embedding $f_m$, appends it to $f_r$, and feeds it into the transformer consisting of three attention blocks. The learned $f_m$ is then projected to $\hat{f}_m$ using a learnable matrix $M_s$, resulting in the style feature $f_s$. Note that $f_m$ extracts information from $f_r$ and can adapt to $f_r$ with a flexible length (referring to our later multiple references). By concatenating $f_t$ and $f_s$ as the condition c in Eq. 1 ($c = [f_t, f_s]$), we can generate stylized images with SD.

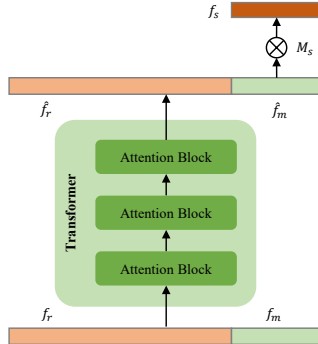

Figure 2: **Structure of StyEmb**.

As shown in Figure 3 (c), this vanilla approach can achieve a desirable stylization effect. However, it reveals two major challenges: *1) the prompt loses controllability over the generated content, and 2) the generated image inherits both the semantic and style features of the style reference images, compromising its content fidelity.*

By further analyzing the results of the original SD and our vanilla StyleAdapter, we get an observation.

**Observation 1: Simply combining the features of the prompt and style reference to guide the generation of images potentially results in a loss of prompt controllability over the generated content.** Figure 3 (b) shows that the original SD generates natural images confirming to the prompt content, e.g. the motorcycle and dog. However, when adapting it to stylized image generation with our vanilla StyleAdapter, the prompts lose their controllability over the generated content, and the content in the style reference becomes dominant, as seen in (c), where the girl from (a) becomes the main object. We explore the insight reason by plotting the attention weights for "motorcycle", "dog", and style features in each cross-attention layer of SD or our vanilla StyleAdapter. Statistic results in (e) reveal that the attention to "motorcycle" and "dog" decreases when involving style features to guide the image generation, while style features gain higher attention. This suggests that simply combining the features of the prompt and style reference makes it difficult to properly utilize these two information sources during stylized image generation. To address this issue, we employ a two-path cross-attention module (TPCA, detailed in 3.3.1) to process these sources separately. Corresponding results in (d) demonstrate that prompts regain controllability over the generated content.

Nonetheless, the second challenge remains unresolved. Results in Figure 3 (d) and Figure 4 (b) indicate that both the content specified in the prompt and style reference appears in the generated images, such as the robot body and natural human face in Figure 4 (b). This issue primarily stems from the tight coupling between semantic and style information in the style reference, leading to another insightful observation.

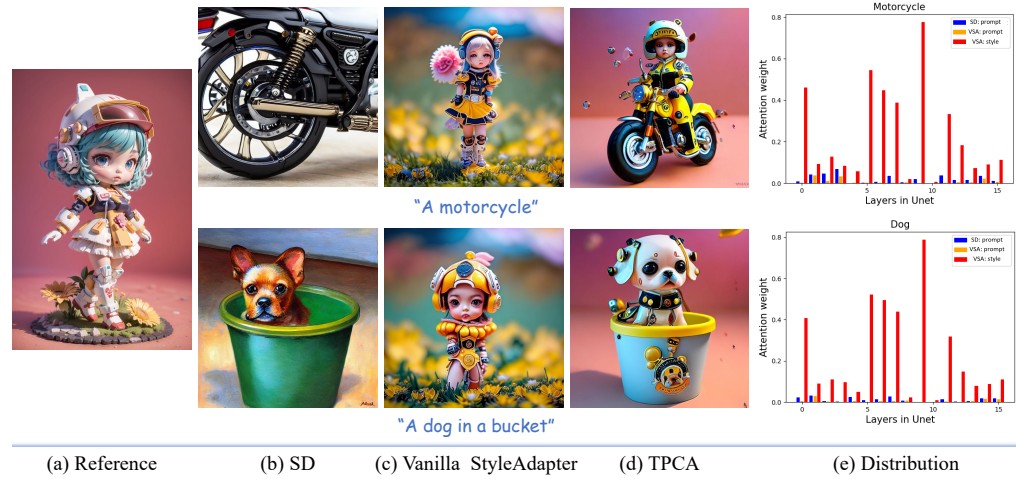

(a) Reference     (b) SD     (c) Vanilla StyleAdapter     (d) TPCA     (e) Distribution

Figure 3: **Illustration of prompt controllability loss.** Without style reference, SD (Rombach et al., 2022) generates images matching content prompts, such as the motorcycle and dog in (b). However, Vanilla StyleAdapter (VSA) concatenates style reference features with prompts, resulting in images dominated by the girl and flowers in the style image, as shown in (c). (e) is the attention weights of keywords (motorcycle and dog) in SD and VSA, which reveal that after combining prompt with style features, VSA reduces prompt attention and focuses more on style features. We propose a two-path cross-attention module (TPCA) to inject prompt and style reference features into the generated images separately, preserving both content and style, as shown in (d).

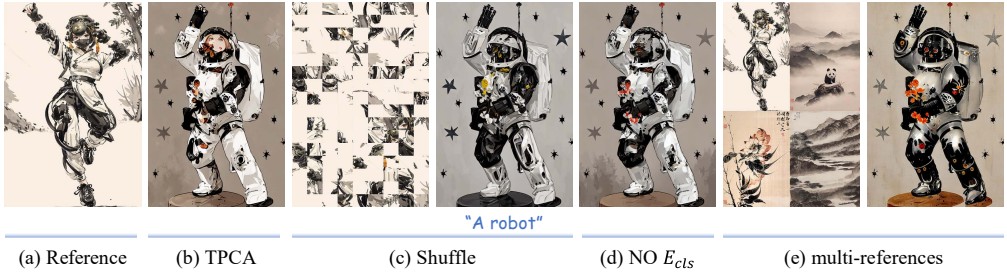

(a) Reference     (b) TPCA     (c) Shuffle     (d) NO $E_{cls}$     (e) multi-references

Figure 4: **Preliminary experimental results on the issue of semantic and style coupling in the style image**. (b) shows a result of our TPCA. It is a robot whose style is similar to the reference but with a human face, due to the tight coupling between the semantic and style information in the reference. Our preliminary experiments suggest that patch-wisely shuffling the reference image (c), removing the class embedding $E_{cls}$ (d) in Eq. 3, and providing multiple diverse reference images (e) can help mitigate this issue.

**Observation 2: Semantic suppressing is required when extracting style features from style references.** Considering that the VM described in extracts style features patch-wise and its class embedding $E_{cls}$ has been proven to be rich in semantic information for classification Dosovitskiy et al. (2021), we aim to shuffle these patches and remove $E_{cls}$ to disrupt and reduce the semantic information in style references. Corresponding generated results are in Figure 4, which successfully replace the natural human face with a robot face. Moreover, the result in (e) suggests that using multiple style images with diverse semantics (e.g., human, panda, flower, and mountain) and similar styles (e.g., ink style) enables the generation model to extract similar style information and disregard their diverse semantic information. These phenomena inspire us to propose a semantic suppressing vision model with multiple style references to obtain semantic-suppressed style features for stylized image generation.

### 3.3 STYLEADAPTER

Motivated by previous observations and analyses, we propose our delicate StyleAdapter, deployed with a two-path cross-attention module (TPCA) to separately process style information and textural prompt, which cooperate with a semantic suppressing vision model (SSVM) to suppress the semantic content of style images.

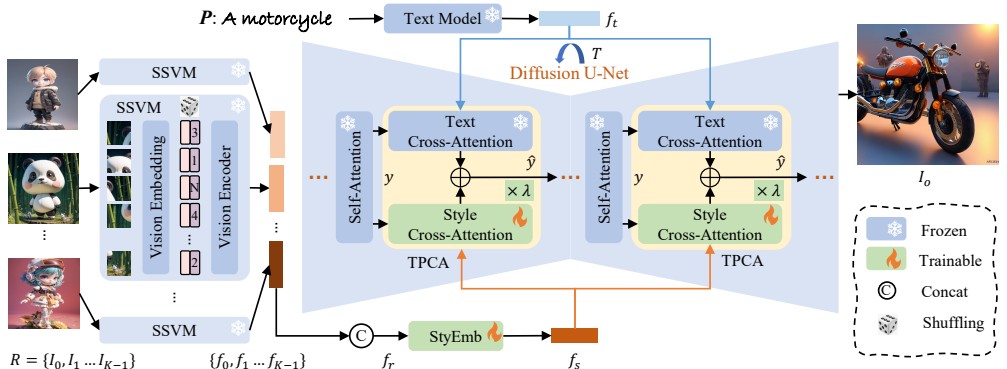

Figure 5: **StyleAdapter Framework.** StyleAdapter is built upon SD (Rombach et al., 2022) and utilizes CLIP's (Radford et al., 2021) text model to extract the features of prompt $P$. It employs a semantic suppressing vision model (SSVM) to extract style information from multiple style reference $R$, and suppresses their semantic information by shuffling the patch-based vision embeddings and removing the original class embedding. Then, the reference features are concatenated as $f_r$ and processed by the StyEmb Module to obtain style feature $f_s$. The prompt feature $f_t$ and style feature $f_s$ are separately processed using the two-path cross-attention module (TPCA) before fusing with learnable coefficient $\lambda$. The fused result is passed to the subsequent SD block. After $T$ sampling steps, StyleAdapter generates a stylized image with content matching the prompt and style conforming to the references.

Specifically, as depicted in Figure 5, our StyleAdapter is based on SD, with conditions comprising a natural language prompt **P** and style reference images $\mathbf{R} = \{I_0, I_1, \ldots, I_{K-1}\}$. The textual feature $f_t$ is extracted using a traditional text model Radford et al. (2021), while the style features $\{f_0, f_1, \ldots, f_{K-1}\}$ are extracted using our proposed SSVM. These style features are processed into $f_s$ using the style embedding module (**StyEmb**). Subsequently, $f_t$ and $f_s$ are independently incorporated into the generation process using our proposed TPCA, before being combined with a learnable weight $\lambda$. The fused result is passed to the subsequent SD block. After $T$ sampling steps, we generate image $I_o$, conforming to the desired content and style. StyleAdapter is learned with $L_{LDM}$ (Eq. 1), where condition $c$ consists of $f_t$ and $f_s$.

### 3.3.1 TOW-PATH CROSS-ATTENTION MODULE

We deploy our two-path cross-attention module after each self-attention module in the diffusion Unet (Ronneberger et al., 2015) model. It consists of two parallel cross-attention modules: text cross-attention and style cross-attention, which are responsible for handling the prompt-based condition and the style-based condition, respectively. The query of both cross-attention modules comes from the spatial feature $y$ of SD. However, the key and value of text cross-attention come from the text feature $f_t$, while the key and value of style cross-attention come from the style feature $f_s$. The attention output of text cross-attention $Attention(\mathbf{Q}_t, \mathbf{K}_t, \mathbf{V}_t)$ has the same formula as Eq. 2 and the output of style cross-attention $Attention(\mathbf{Q}_s, \mathbf{K}_s, \mathbf{V}_s)$ can be formulated as:

$$\begin{cases} \mathbf{Q}_s = \mathbf{W}_{Qs} \cdot y; \ \mathbf{K}_s = \mathbf{W}_{Ks} \cdot f_s; \ \mathbf{V}_s = \mathbf{W}_{Vs} \cdot f_s; \\ Attention(\mathbf{Q}_s, \mathbf{K}_s, \mathbf{V}_s) = softmax(\frac{\mathbf{Q}_s \mathbf{K}_s^T}{\sqrt{d}}) \cdot \mathbf{V}_s. \end{cases} \quad (4)$$

The outputs of these two attention modules are then added back to y and fused with a learnable parameter $\lambda$. This produces a new spatial feature $\hat{y}$ that is fed to the subsequent blocks of SD. The process can be expressed as:

$$\hat{y} = Attention(\mathbf{Q}_t, \mathbf{K}_t, \mathbf{V}_t) + \lambda Attention(\mathbf{Q}_s, \mathbf{K}_s, \mathbf{V}_s). \quad (5)$$

It is worth noting that since SD already has a strong representation for the prompt, we retain the original cross-attention in SD as our text cross-attention and freeze it during training. In contrast, style cross-attention is implemented with the same structure as text cross-attention and is trained to adapt to the style reference.

### 3.3.2 SEMANTIC SUPPRESSING VISION MODEL

Our semantic suppressing vision model (SSVM) aims to suppress the semantic information in style references while extracting style features, mitigating the negative impact on the generated images. It

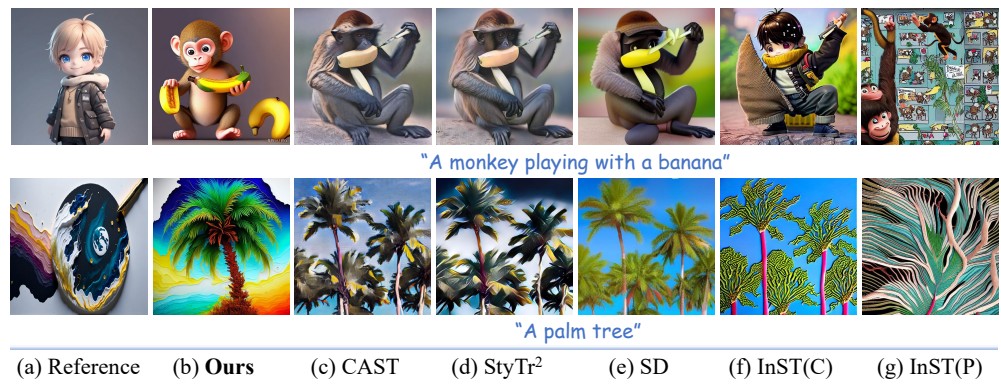

"A monkey playing with a banana"

"A palm tree"

| (a) Reference | (b) **Ours** | (c) CAST | (d) StyTr$^2$ | (e) SD | (f) InST(C) | (g) InST(P) |

Figure 6: Qualitative comparison with state-of-the-art methods using a single style reference image: Traditional methods like CAST (Zhang et al., 2022c) and StyTr$^2$ (Deng et al., 2022) focus on color transfer, whereas diffusion-based methods like SD (Rombach et al., 2022) and InST (Zhang et al., 2022a) struggle with content-style balance. Our StyleAdapter captures more style details from references, such as brushstrokes and textures, while better matching prompt content.

achieves semantic suppressing from three aspects. First, it removes $E_{cls}$ in Eq. 3, which is rich in semantic information. Second, it patch-wisely shuffles the reference image by randomly shuffling the $E_{pos}$ in Eq. 3 before adding them to the patch embeddings. Third, it adopts multiple semantic-diverse style images as references.

Specifically, SSVM parallelly processes multiple style references $R = \{I_0, I_1, \ldots, I_{K-1}\}$. $K$ denotes the number of style reference images, and $K = 3$ while training, although it can be any positive integer. It patch-wisely extracts the vision embeddings with a vision embedding module, and directly adds the vision embeddings with randomly shuffled position embeddings ($E_{pos}$). Then added results are sent into a vision encoder to attain the vision features of each reference.

## 4 EXPERIMENTS

### 4.1 EXPERIMENTAL SETTINGS

**Datasets.** We employ a subset of the LAION-AESTHETICS (Schuhmann et al., 2022) dataset, containing 600K image-text pairs, for training. **While training, the style references are the augmentation results attained from the image in the text-image pairs in the LAION-AESTHETICS.** To evaluate our method, we construct a diverse testset comprising 50 prompts, 50 content images, and 8 groups of style references. Therefore, there are a total of 400 test pairs. More details are in Appendix A and Figure 9.

**Implementation Details.** We adopt the SD model (Rombach et al., 2022) (version 1.5) as our base and use CLIP's (Radford et al., 2021) text and vision encoders, implemented with a large ViT (Dosovitskiy et al., 2021) (patch size 14). We fix the original SD and CLIP parameters, updating only StyEmb and style cross-attention module weights. We use Adam (Kingma & Ba, 2014) optimizer with a learning rate of $8 \times 10^{-6}$ and batch size of 8. Experiments run on 8 NVIDIA Tesla 32G-V100 GPUs. Input and style images are resized to $512 \times 512$ and $224 \times 224$, respectively. Data augmentation includes random crop, resize, horizontal flipping, rotation, etc., generating $K = 3$ style references per input image during training (with variable $K$ at inference). We set sampling step $T = 50$ for inference.

**Evaluation metrics.** This paper evaluates generated images both **subjectively** and **objectively** in terms of text similarity, style similarity, and quality. We conduct a **User Study** for subjective assessment and employ a CLIP-based (Radford et al., 2021) metric to objectively measure text similarity (Text-Sim) and style similarity (Style-Sim) using cosine similarity. Additionally, we utilize FID (Seitzer, 2020) to assess image quality.

### 4.2 COMPARISONS WITH STATE-OF-THE-ART METHODS

In this section, we conduct comparisons with current state-of-the-art related methods, including two traditional style transfer methods: CAST (Zhang et al., 2022c) and StyTr$^2$, three SD-based methods: InST (Zhang et al., 2022a), Textual Inversion (TI) (Gal et al., 2022), and LoRA (Hu et al., 2021), and

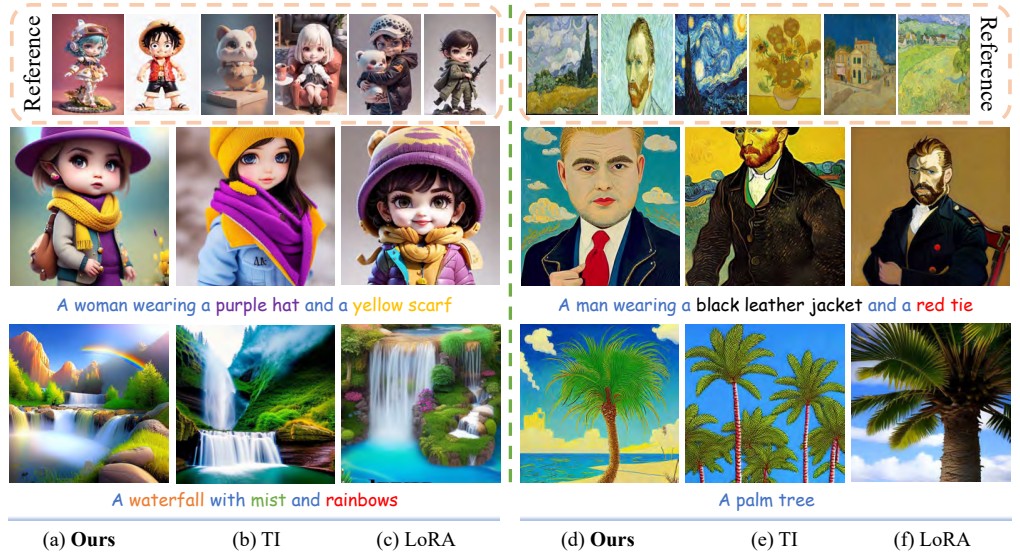

Figure 7: Qualitative comparison with TI (Gal et al., 2022) and LoRA (Hu et al., 2021) using multiple style reference images. TI and LoRA, trained on references, perform well in stylization but show insensitivity to the prompt. Conversely, our StyleAdapter, without requiring test-time fine-tuning, performs better in generating both style and content.

SD (Rombach et al., 2022) itself. Note that we use the image-to-image mode in SD (Rombach et al., 2022) to generate a stylized image from the content image in the testset and the prompt generated from the reference image with BLIP2 (Li et al., 2023).

**Comparisons based on single style reference.** We compare our method with state-of-the-art single style reference methods in Figure 6. CAST (Zhang et al., 2022c) and StyTr$^2$ (Deng et al., 2022) perform relatively coarse-grained color transfer. SD (Rombach et al., 2022) yields unsatisfactory stylization due to poor text representation from reference images. InST (Zhang et al., 2022a), based on textural inversion, generates stylized images using content image (InST(C)) or prompt (InST(P)). InST(C) outperforms previous methods and InST(P) in stylization, but its content is dominated by the style reference image (it generates a boy rather than the monkey indicated in the prompt in the first sample), or unnatural texture results in strange appearances (the result of the second sample). InST(P) generates content closer to the prompt but with different styles. On the contrary, our method generates images faithful to style reference and prompt content. Table 1 shows the quantitative evaluation, indicating our method achieves a better balance between prompt fidelity, style fidelity, and image quality.

**Comparisons based on multiple style reference.** Unlike our method, which is a unified model that can be generalized to different styles without test-time fine-tuning, TI (Gal et al., 2022) and LoRA (Hu et al., 2021) require training on the style reference images for each style. Figure 7 and Table 1 present the visual and quantitative results, respectively. TI (Gal et al., 2022) inverses the style references into a learnable textural embedding embedded into the prompt for guiding the generation. It performs better in style similarity (high score in Style-Sim). However, Its generated content cannot match the prompt accurately, such as the purple hat, yellow scarf, red tie, and rainbows, indicated in the prompts but missed in their corresponding generated results of TI (Gal et al., 2022), leading to a lower score in Style-Text. Our proposed method is comparable to LoRA (Hu et al., 2021) in style, but it performs better in text similarity, according to the higher score of Text-Sim and the generated tie and rainbows responding to the prompts in the visualized results, which demonstrates that our StyleAdapter achieves a better balance between content similarity, style similarity, and generated quality in objective metrics.

**User Study.** We conducted a user study for a comprehensive evaluation, selecting 35 generated results across all styles and involving 24 professional users in AIGC to assess text similarity, style similarity, and quality. With 2520 votes, Table 1 shows our StyleAdapter's results are preferred in all three aspects. We observe that the difference between objective and subjective metrics may arise from users considering all aspects jointly while making decisions, while the objective metrics assess them independently. This indicates our results achieve a better trade-off between quality, text similarity, and style similarity.

Table 1: **Objective** and **Subjective** quantitative comparisons with the state-of-the-art methods. Our proposed method achieves a better balance in text similarly, style similarity, and quality, and attains more preference from expert users.

| Methods | Single-reference | | | | | | Multi-reference | | | | User Study | | | |
|---|---|---|---|---|---|---|---|---|---|---|---|---|---|---|
| | CAST | StyTr$^2$ | InST(P) | InST(C) | SD | **Ours** | TI | LoRA | **Ours** | | CAST | InST(P) | LoRA | **Ours** |
| Text-Sim ↑ | 0.2323 | 0.2340 | 0.2204 | 0.1682 | 0.2145 | 0.2435 | 0.1492 | 0.2390 | 0.2448 | Text-Sim ↑ | 0.2310 | 0.0548 | 0.2869 | **0.4274** |
| Style-Sim ↑ | 0.8517 | 0.8493 | 0.8616 | 0.8707 | 0.8528 | 0.8645 | 0.9289 | 0.9034 | 0.9031 | Style-Sim ↑ | 0.3857 | 0.0286 | 0.1881 | **0.3976** |
| FID ↓ | 163.77 | 151.45 | 177.91 | 153.45 | 189.34 | 141.78 | 139.56 | 137.40 | 140.97 | Quality ↑ | 0.2071 | 0.0452 | 0.3238 | **0.4238** |

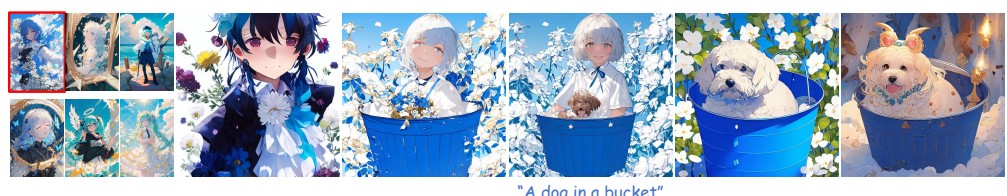

"A dog in a bucket"

(a) References  (b) Vanilla StyleAdapter  (c) TPCA  (d) TPCA - $E_{cls}$  (e) TPCA - $E_{cls}$ + Shuffle  (f) StyleAdapter

Figure 8: Ablation study. Results (b)∼(e) are attained with a single reference in the red box while (f) is attained with all the references. Detailed explanations are in 4.3.

**Cooperation with existing adapters.** Our StyleAdapter can cooperate with existing adapters, such as T2I-adapter (Mou et al., 2023). Results in the last column of Figure 1 and Figure 11 in Appendix D show that with the guidance of the additional sketches, the shape of the generated contents is more controllable.

### 4.3 ABLATION STUDIES

We evaluate the effectiveness of our TPCA module and SSVM through experiments, with qualitative results in Figure 8. Using Vanilla StyleAdapter (VSA) to fuse the prompt and single style reference information (the reference in red box), as in (b), the content is dominated by the reference girl, ignoring the prompt's dog and bucket. To improve prompt controllability, we process these sources separately using TPCA, resulting in (c), where the bucket appears but the dog is missing and the reference girl remains. This issue stems from the tight coupling between semantic and style information in the style reference. We remove the class embedding $E_{cls}$ via SSVM, as in (d), generating the dog and bucket but still including the reference girl. Further shuffling the patches in SSVM, as in (e), disrupts semantic and style information coupling, removing the reference girl and emphasizing the prompt's dog in the bucket, but with less similar style to the reference. Employing multiple style references, as in (f), results in content dominated by the prompt and style closely resembling the reference images. These outcomes demonstrate the effectiveness of our TPCA module and SSVM. The quantitative results and more discussions of our StyleAdapter are in Appendix B and Appendix C.

### 4.4 LIMITATIONS AND BORDER IMPACT

Our work aims to propose a unified model for different styles without test-time fine-tuning. Compared to LoRA (Hu et al., 2021), which trains a specific model for each style, our model may not always achieve the stylization performance of LoRA. Further improving the generated quality and generalization of StyleAdapter is part of our ongoing work.

Since our model primarily relies on a pre-trained stable diffusion, the generated data is constrained by the dataset distribution used for training the stable diffusion. Consequently, this could result in the generation of unethical images, raising concerns about content quality and ethical considerations.

## 5 CONCLUSION

In this paper, we propose StyleAdapter, a unified stylized image generation model capable of producing a variety of stylized images that match both the content of a given prompt and the style of reference images, without the need for test-time fine-tuning. It introduces a two-path cross-attention (TPCA) module to separately process style information and textual prompt, which cooperate with a semantic suppressing vision model (SSVM) to suppress the semantic content of style images. This design is motivated by our in-depth observations and analyses. TPCA ensures the controllability of the prompt over the content of the generated images while SSVM mitigates the negative impact of semantic information in style references, and finally attains high-quality stylized images that conform to both the prompt and style references.

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

## A  MORE DETAILS OF TESTSET

To evaluate the effectiveness of our proposed method, we construct a testset that consists of prompts, content images, and style references. **Prompts**: We use ChatGPT (cha) to generate diverse prompts and manually filter out the irrelevant or low-quality ones. The final testset contains 50 prompts which are listed on the right of Figure 9. **Content images**: To meet the requirement of the content-based methods, such as CAST (Zhang et al., 2022c) and StyTR$^2$ (Deng et al., 2022), and align to our proposed method that prompts determine the content of the generated image with SD, we use SD to generate the content images from the prompts in the test set. In this paper, we fix the seed to 993 for generating the content images. **Style references**: We collect 8 sets of style references from the Internet[1], each containing 5 to 14 images. We use them as our multi-reference inputs and they are shown on the right of Figure 9.

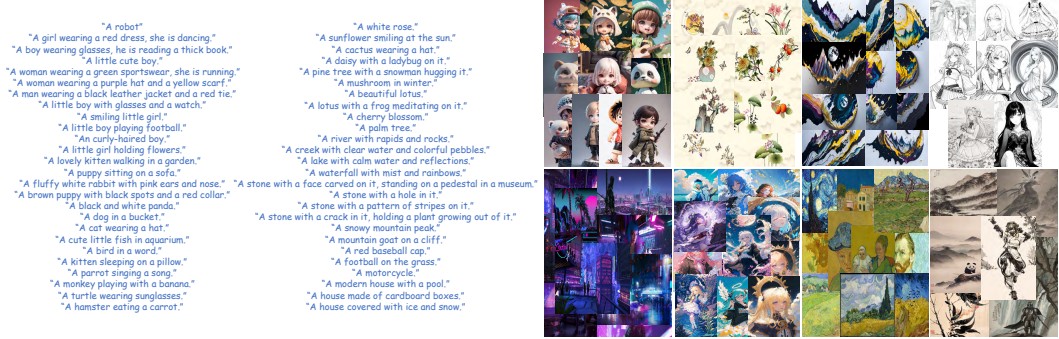

Figure 9: **Details of Testset.** Sentences on the left are 50 prompts that we used in this work, and images on the right are 8 sets of style references that we collect from the Internet.

## B  MORE DISCUSSIONS ABOUT ABLATION STUDY

### B.1  QUANTITATIVE RESULTS OF ABLATION STUDY

Table 2 presents the quantitative results of our ablation study. Compared to Vanilla StyleAdapter (VSA), our StyleAdapter that uses two-path cross-attention modules (TPCA) achieves higher scores in

---

[1]The style references are collected from https://civitai.com, https://wall.alphacoders.com, and https://foreverclassicgames.com.

terms of Text-Sim, which means its results are more consistent with the prompts, although sacrificing some performance of stylization (as indicated by the lower score in terms of Style-Sim). To further suppress the semantic information in the style references while extracting style information, we employ a semantic suppressing vision model (SSVM) to extract style features. By removing $E_{cls}$, SSVM can slightly improvement of the score of Text-Sim while barely affecting the performance of stylization. Further adopting patch-wise shuffling significantly suppresses the semantic information in the style references and boosts the score of Text-Sim by about 0.0326. However, it also degrades the style of the generated results considerably, as shown by the large drop in the score of Style-Sim. By further taking multiple references as input, our StyleAdapter enhances both Text-Sim and Style-Sim, achieving a better balance between the content and style of the generated results. Moreover, our TPCA and SSVM enhance the quality of generated images, as indicated by the lower FID score.

Table 2: **Quantitative results of ablation study.** Our method based on TPCA achieves a significant improvement in Text-Sim compared to VSA. Employing strategies in SSVM can progressively improve Text-Sim, and eventually attain a better balance between Text-Sim and Style-Sim after utilizing multiple references. Moreover, our TPCA and SSVM enhance the quality of generated images, as indicated by the lower FID score.

| VSA | TPCA | No $E_{cls}$ | Shuffling | multi-reference | Text-Sim ↑ | Style-Sim ↑ | FID ↓ |
|---|---|---|---|---|---|---|---|
| ✓ | | | | | 0.1263 | 0.9362 | 186.17 |
| | ✓ | | | | 0.2089 | 0.8963 | 145.37 |
| | ✓ | ✓ | | | 0.2109 | 0.8921 | 141.99 |
| | ✓ | ✓ | ✓ | | 0.2435 | 0.8645 | 141.78 |
| | ✓ | ✓ | ✓ | ✓ | 0.2448 | 0.9031 | 140.97 |

## B.2 ADAPTIVE $\lambda$

As defined in Eq. 5 in the paper, our proposed two-path cross-attention modules fuse the information of the prompt and style references with $\lambda$. $\lambda$ is an adaptive parameter that controls the trade-off between the content from the prompt and the style from the references. As shown in Figure 10, when we scale down $\lambda$ by a factor smaller than 1.0, the style features from the references fade away gradually, and the generated images become more natural. On the other hand, when we scale up $\lambda$ by a factor larger than 1.0, the style features in the generated images become more prominent, such as the 3D shape and fantastic appearance. However, the dog also loses its natural look. Therefore, users can customize the generated results according to their preferences by adjusting $\lambda$. The results shown in this paper are obtained with the original $\lambda$ without any scaling factor unless otherwise stated.

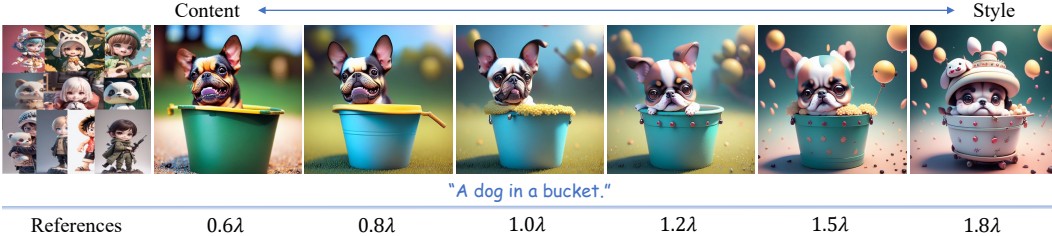

Figure 10: **Adaptation of $\lambda$.** By tuning $\lambda$ with an appropriate factor, we can obtain a generated image with a better balance between the content from the prompt and the style from the references. Factors smaller than 1.0 tend to suppress style features and produce a more natural image, while factors larger than 1.0 tend to enhance style features.

## C DISCUSSION ABOUT MODEL SIZE

Apart from the SD (Rombach et al., 2022) model, the text and vision models borrowed from CLIP (Radford et al., 2021) are also used to extract the features of the prompt and style images in the previous related works, such as InST (Zhang et al., 2022a) and LoRA (Hu et al., 2021). Therefore,

the only novel modules in this work are the Style Emb and TPCA modules. Their model sizes are $148M$ and $168M$, respectively. Although the model sizes of InST (Zhang et al., 2022a) ($15M$) and LoRA (Hu et al., 2021) ($37M$) are smaller, an InST (Zhang et al., 2022a) model can only process the style from a specific image that is used during training, while a LoRA (Hu et al., 2021) model is only suitable for a certain kind of styles. In contrast, our model can handle various styles by taking different style references at inference time.

## D    MORE GENERATED RESULTS

Figure 11 shows more generated results. Given multiple style reference images, our StyleAdapter can generate images that adhere to both the style and the prompts in a single pass. For example, the first two generated results in the first row are a panda and a woman wearing a purple hat and a yellow scarf, which are consistent with their prompts, respectively. Both of them have a 3D shape and a cute look, which are similar to their style references in the first column. Moreover, our StyleAdapter can cooperate with the existing controllable synthesis methods, such as T2I-Adapter (Mou et al., 2023), to generate high-quality images more controllably and stably. For example, given sketches (attached in the corner of the generated results in the fourth column), our method can generate objects following the sketches but with the style of the reference images.

Besides, we also evaluate our StyleAdapter with more styles. Results are in Figure 12. We can see that our model performs well on different styles without test-time fine-tuning.

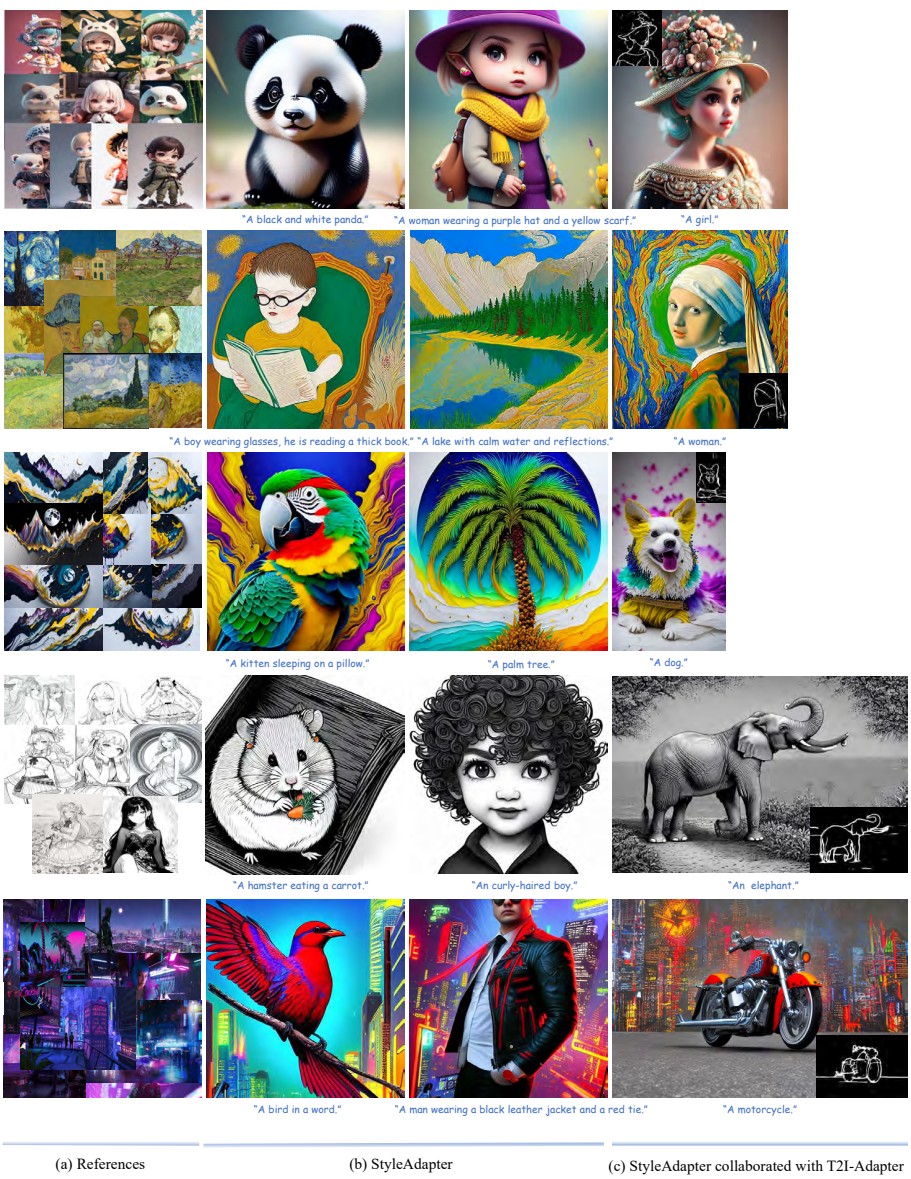

(a) References        (b) StyleAdapter        (c) StyleAdapter collaborated with T2I-Adapter

Figure 11: **More generated results.** Given multiple style reference images, our StyleAdapter can generate images that adhere to both style and prompts in a single pass. Moreover, our method shows compatibility with additional controllable conditions, such as sketches.

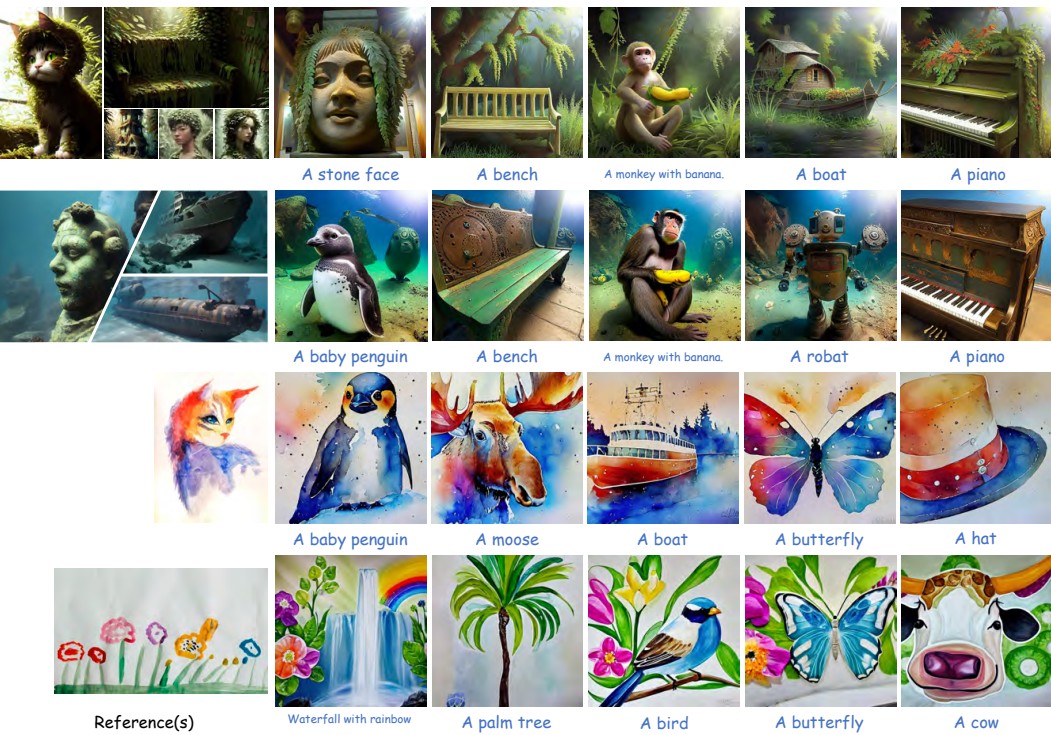

Figure 12: **Evaluations in additional styles**. Our model performs well on various styles without test-time fine-tuning, including the last two single references provided by StyleDrop (Sohn et al., 2023) (StyleDrop requires training on each style reference and further fine-tuning with human feedback).

