# OpenReview forum: "StyleAdapter: A Unified Stylized Image Generation Model without Test-Time Fine-Tuning"
_ICLR.cc/2024/Conference — ICLR 2024 Conference Withdrawn Submission_

### Official Review · Reviewer_Civx · 2023-10-30

**Soundness:** 2 fair
**Presentation:** 3 good
**Contribution:** 2 fair
**Rating:** 3
**Confidence:** 4

**Summary:**

The paper proposed a StyleAdapter to improve the generation performance with both text prompt and style image input without test-time fine-tuning. The proposed two-path cross-attention module improve the controllability of the prompt over the generated content while mitigating transferring the non-style information from the style references. The work is novel and the results shows some improvement over the vanilla diffusion model.

**Strengths:**

1. The proposed approach is novel and tackles challenging problem of controlling the generation content.
2. The proposed approach is straightforward. The approach diagram is clear and easy to follow.

**Weaknesses:**

1. The qualitative results in Figure 6 are confusing. The stylized image from the proposed approach does not follow the style well.
2. The evaluation is insufficient and confusing. The results in Table 1 shows there is no significant improvement between the proposed approach and the baselines. The proposed approach is trying to achieve good alignment for both text prompt and the style. However, the paper evaluates the text alignment, style alignment, and image quality separately. It is hard to see how much improvement the proposed approach achieved compared to the other approaches.
3. More ablation studies are needed to justify the contribution. For example, how much does image shuffling improve the performance? How does the combined attention impact the attention weight in Figure 3(e). How would the design of the transformer model in Figure 2 impact the performance?
4. Would mixing various styles impact the performance. How to keep the extracted style consistent between multiple images?
5.  The paper writing style can be improved. For example, the vanilla style adapter is not well defined in section 3.2. Figure 2 is not aligned with the text description. There are also a lot of typos in the paper, e.g. TOW, etc.

**Questions:**

The extra questions besides the questions in weakness:
6. How does the proposed approach disentangle the style embedding with the text prompt embedding? What will happen if the text prompt include some style requirement?

---

### Official Review · Reviewer_scyk · 2023-10-30

**Soundness:** 3 good
**Presentation:** 3 good
**Contribution:** 2 fair
**Rating:** 5
**Confidence:** 3

**Summary:**

This paper proposes a LoRA-free method to generate stylized images, which enables one-shot and few-shot stylization in an unified model. The key idea is to introduce a two-path cross-attention module that fuses the content information from text prompt and style information from style images, constructing a text-image condition to guide the Stable Diffusion to generate image. Furthermore, a semantic suppressing vision model is used to suppress the semantic content of style images.Experiments show that the proposed method outperforms some training-free works.

**Strengths:**

This paper is clear in its writing and easy to follow. The proposed approach is reasonable. There are sufficient experiments to demonstrate the effectiveness of each part in the method.

**Weaknesses:**

1. It's not novel to use an extra cross-attention to inject image condition into the T2I model, as the similar idea has been proposed before. This paper seems like a straightforward application of this idea to stylization.

2. The decoupling strategies are also simple and like some engineering tricks, which may limit the quality of generated image. This paper lacks an in-depth discussion of how to disentangle the style information and inject it into the T2I model.

3. The method lacks a lot of details, especially in  embedding suppressing and multiple style references. More implementation details should be included to ensure reproducibility.

4. limited styles are provided for test to fully evaluate the method.

**Questions:**

Please check weakness.

---

### Official Review · Reviewer_tsHj · 2023-10-31

**Soundness:** 3 good
**Presentation:** 3 good
**Contribution:** 2 fair
**Rating:** 5
**Confidence:** 5

**Summary:**

The paper presents a StyleAdapter, fine-tuning free style-adaptive text-to-image generation based on a few style reference images. There are two technical innovations. First is the semantic-suppressing visual model (SSVM), which translates a set of style reference images into style embedding. To suppress the semantics while extracting style, the paper proposes to shuffle patches. Second is two-path cross-attention. In addition to existing (thus frozen) pre-trained text cross-attention module, the paper proposes to train a cross-attention with respect to the style embedding and sum outputs of two cross-attention modules. SSVM and Stable diffusion are frozen, and style cross-attention module and style embedding module are trained on a 600K image-text paired data (LAION aesthetics). Experimental results show that the model can generate images consistent in styles to style reference images.

**Strengths:**

- The paper is well written and easy to follow.

- The paper tackles an important problem of style personalization of text-to-image models. Compared to some existing works, the proposed method is faster.

- The use of shuffling the suppress semantic information is a good idea, though it may limit the definition of style the model can handle (see Weaknesses section for more discussion).

- The paper conducted a comprehensive study (e.g., automated scoring based on CLIP and user study).

**Weaknesses:**

- Overall technical contribution is rather marginal.

- While the proposed method seems to outperform other diffusion-based style consistent image generation, it is unclear how it compares against results based on the StyleDrop. Results in Figure 12 suggests that, while generated images are consistent styles to each other, it is not as consistent to the style reference images to StyleDrop.

- First two rows of Figure 12 suggest that the style is not really well transferred to different text prompt. For example, baby penguin, monkey are not really "rusty" for the second row, and monkey or bench are not covered with grass in the first row.

- Some styles considered in the paper may be too generic and may be generated via text prompt engineering. For example, Figure 11 third row. Also, the generation results only captures the texture, but not the color scheme or the layout. It would be great if authors can provide more results on the diverse set of style reference images (e.g., those considered in StyleDrop) and discuss the limitation more comprehensively.

- It would be great to include the interface used for user study and more details about how the question was asked, which options raters are provided to answer, etc.

- This begs a question on the definition of "style" the paper has considered. Are color or texture schemes style? How about the layout or composition? It seems the shuffling technique of SSVM seems to make style to be limited to more local properties (e.g., color, texture) but not really the global properties. It would be great if authors can provide textual description of the style that the model is trying to extract. Also, it would be great to have an ablation study with different shuffling strategy (e.g., block shuffling to retain more global properties).

**Questions:**

Please see weaknesses.

---

### Official Review · Reviewer_6r1g · 2023-11-01

**Soundness:** 3 good
**Presentation:** 3 good
**Contribution:** 2 fair
**Rating:** 5
**Confidence:** 5

**Summary:**

Problem statement
* Text prompts in T2I methods are not expressive enough to specify styles.
* Textual inversion reflects styles of reference images by projecting reference images to text embedding.
  * It loses rich details in the references due to modality mismatch between vision and language.
* DreamBooth and LoRA employ extra per-style training.
  * It is time-consuming.
* Using image encoder of CLIP in text-conditioning part loses text control in content and accompanies content leak from the style reference.

This paper tackles generating images:
* Their content is specified by text prompts.
* Their style is specified by reference images.
* The proposed method does not require per-style finetuning.
* It supports (optional) conditions using T2I-adapter or ControlNet.

Two-path cross-attention module
* It separately process style (reference images) and content (text prompt).
* The two cross-attention outputs are linearly added.

Semantic suppressing vision model
* It extracts style without content from the reference images.

Experiments
* Competitors: style transfer methods, textual inversion, dreambooth, lora

**Strengths:**

1. Building the motivation from the vanilla style adapter is a nice try.

1. Logical ground for semantic suppression is sound.

1. The proposed method faithfully reflect the text prompts and style reference images in the results. User study supports it and I agree.

1. Ablation study on TPCA, $E_{cls}$, shuffle are sound.

**Weaknesses:**

1. I am not sure if the vanilla style adapter is a proper starting point. Is it better than simply using CLIP image encoder by itself?

2. *Test-time* finetuning is a misleading word choice because existing methods do not require finetuning everytime it generates images with different prompts but require finetuning for different styles. Perhaps per-style finetuning?

3. DreamBooth and LoRA takes only minutes for finetuning for given small number of images as in the configuration of this paper. It can be both practical and impractical regarding the application. It would be clearer to specify the target application.

4. Figure 3 should include below to support the motivation.
* Attention weight distribution of TPCA
* Diverse examples with different prompts or [same prompts, different noises],  maybe in the appendix

5. Configuration for the comparison to existing methods should be clear.
* Results from the existing methods are there. However, inputs for them are not clearly described. Hence, currently the comparisons are invalid.
* I am not sure how style transfer methods can receive content images equivalent to the text prompts.
* Prompts for textual inversion also should be described. Probably it is "A palm tree in style of S*".

6. Two-path cross-attention is not novel.
* But I cannot use this for the reason for rejection because the references are not accepted anywhere. It will be a reason if someone knows an accepted paper.
* Mou et al, T2I-Adapter: Learning Adapters to Dig out More Controllable Ability for Text-to-Image Diffusion Models
* Ye et al., IP-Adapter: Text Compatible Image Prompt Adapter for Text-to-Image Diffusion Models

(minor)

7. Unnecessary abbreviations harm readability, e.g., VM for vision model.

8. The target dimension of the concatenation operation in $3.2 is not specified. I suppose it is concatenation of the tokens.

9. Caption of Figure 4 lacks clarity. I suppose (c,d,e) all use TPCA, implied from the text.

10. AIGC is not defined.

11. Using boldface for different categories (single-ref and multi-ref) may improve readability.

12. Reference image set for FID is not defined.

**Questions:**

The authors consider the proposed method as a *unified* stylized image generation model. I wonder about the context of unification. What does it unify? If it is not a unification of something, I suggest removing it because it is misleading.

Please refer to Weaknesses for potential adjustments on my rating.